# Nanomaterial-Based Sensor Array Signal Processing and Tuberculosis Classification Using Machine Learning

Chenxi Liu [1], Israel Cohen [1,*], Rotem Vishinkin [2] and Hossam Haick [2]

1. Andrew and Erna Viterbi Faculty of Electrical and Computer Engineering, Technion–Israel Institute of Technology, Haifa 3200003, Israel; mailtoliuchenxi@gmail.com
2. Department of Chemical Engineering and Russell Berrie Nanotechnology Institute, Technion–Israel Institute of Technology, Haifa 3200003, Israel
* Correspondence: icohen@ee.technion.ac.il

**Abstract:** Tuberculosis (TB) has long been recognized as a significant health concern worldwide. Recent advancements in noninvasive wearable devices and machine learning (ML) techniques have enabled rapid and cost-effective testing for the real-time detection of TB. However, small datasets are often encountered in biomedical and chemical engineering domains, which can hinder the success of ML models and result in overfitting issues. To address this challenge, we propose various data preprocessing methods and ML approaches, including long short-term memory (LSTM), convolutional neural network (CNN), Gramian angular field-CNN (GAF-CNN), and multivariate time series with MinCutPool (MT-MinCutPool), for classifying a small TB dataset consisting of multivariate time series (MTS) sensor signals. Our proposed methods are compared with state-of-the-art models commonly used in MTS classification (MTSC) tasks. We find that lightweight models are more appropriate for small-dataset problems. Our experimental results demonstrate that the average performance of our proposed models outperformed the baseline methods in all aspects. Specifically, the GAF-CNN model achieved the highest accuracy of 0.639 and the highest specificity of 0.777, indicating its superior effectiveness for MTSC tasks. Furthermore, our proposed MT-MinCutPool model surpassed the baseline MTPool model in all evaluation metrics, demonstrating its viability for MTSC tasks.

**Keywords:** tuberculosis; multivariate time series classification; sensor signal processing; graph convolution network; Laplacian matrix

## 1. Introduction

Tuberculosis (TB) is an ancient, chronic disease caused by the bacillus Mycobacterium tuberculosis, which threatens an estimated 25% of the world's population, with a 5–10% lifelong risk of developing into TB disease. It usually affects the lungs and can spread from person to person through the air. Pulmonary TB symptoms include a chronic cough, weight loss, chest pain, weakness, fatigue, night sweats, and fever [1]. TB has affected humanity for over 4000 years, and over 10 million people become infected annually. Therefore, it remains one of the leading causes of morbidity and mortality worldwide, especially in developing countries.

Early recognition of TB and prompt detection of drug resistance is critical to reducing its global burden. To overcome this problem, Vishinkin et al. [2] proposed a novel diagnostic pathway to detect TB in a noninvasive, reliable, and rapid manner. They developed a new biomedical apparatus containing a wearable and flexible polymer pouch for collecting and storing TB-specific volatile organic compounds (VOCs) that can be detected and quantified from the air above the skin (the skin's headspace). An abnormal pattern of VOC concentrations that deviates from the healthy pattern may indicate either TB infection or a high risk of infection with TB. The collected VOCs will then be fed into a set of

specially-designed nanomaterial-based sensors capable of detecting a variety of skin-based TB VOCs [3–6]. Finally, the sensors will translate these collected VOCs into the time series of resistance signals. Ultimately, the output multivariate time series (MTS) sensor signals will be used as feature inputs in ML models for the final discrimination between positive TB cases and healthy controls.

Machine learning (ML) has gained much popularity in recent years. Neural networks (NNs) have achieved considerable success in many tasks, such as computer vision (CV), speech, and natural language processing (NLP). The main characteristics that favored the rise of these algorithms are (i) the use of large annotated datasets and (ii) networks with deep structures [7]. However, the first requirement cannot be fulfilled in some natural settings, such as medicine, biology, and chemical engineering, for several reasons. First, the resources can be limited. Obtaining and labeling data can be costly and might take an extended period. Therefore, it is unrealistic to have large datasets under such conditions. Secondly, the standard deeper structure means a model with higher complexity and a more significant number of trainable parameters. This is highly prone to cause overfitting problems and poor results, especially when the trainable samples are limited. Finally, some standard ML systems and deep neural networks are unsuitable for small data settings compared to big data scenarios since insufficient training samples can compromise the learning success [8]. Therefore, learning from a small dataset is highly challenging, and many unresolved problems still need to be solved in small dataset scenarios.

Recent studies have shown that several sub-domains of MLs are trying to solve the small dataset problem from different perspectives [8]. One is by attempting to mitigate the necessity of big training data, such as transfer learning, which aims to learn representations from one domain and then transfer the learned features to a similar and closely related domain [7]. Another approach is using surrogate data, which can be generated from random numbers to imitate the distribution of the original dataset [9]. The first approach is more prevalent in CV and NLP tasks since many large datasets can be used to train the models. The second approach is more common in time series analysis.

The diagnosis of TB is an example of the application of ML to small dataset problems. In this paper, our study builds upon prior work [2] and draws inspiration from existing low-power deep learning models [10,11]. Our primary objective is to develop several low-power ML-based networks to classify the input from nanomaterial-based sensor signals accurately and predict their corresponding labels for diagnosing TB disease. This study also seeks to expand the range of viable options for domains that cannot employ or depend on conventional ML models that typically have high computational resource requirements and power consumption. Finally, our research strives to offer more effective and suitable model options for low-power ML applications in multivariate time series classification (MTSC) tasks or similar domains. Our main contributions can be summarized as follows:

- We utilize multiple data preprocessing techniques, such as sensor signal extraction, data normalization, data calibration, and sensor selection, which apply to similar tasks involving MTS sensor signals.
- We propose various ML-based models, namely long short-term memory (LSTM), convolutional neural network (CNN), Gramian angular field-CNN (GAF-CNN), and multivariate time series with MinCutPool (MT-MinCutPool), to classify the small TB dataset, where the proposed low-power model features a simplified and shallow network architecture, incorporating a limited number of parameters. This design results in lowered computational complexity and effectively reduces power consumption. We then compare the performance of our proposed models with several state-of-the-art methods commonly used in MTSC tasks.
- To encourage further research on MTSC with small-dataset problems, we provide an open-source of our work, which is accessible on 5 March 2023 at: https://github.com/ChenxiLiu6/TB-Classification.git.

The structure of this paper is as follows. In Section 2, we introduce the related work on nanomaterial-based sensors for disease diagnosis by using disease-related VOCs, and

describe the sensors we used and their working mechanisms. In Section 3, we provide background information on MTSC tasks and describe some state-of-the-art approaches for solving MTSC tasks. Next, in Section 4, we present the dataset we use in our study and the data preprocessing methods we employed. Then, in Section 5, we propose four different classification methods, namely LSTM, CNN, GAF-CNN, and MT-MinCutPool, which are appropriate for small TB dataset classification problems. In Section 6, we introduce the evaluation metrics used in this study and the experimental setup for each model. In Section 7, we present the results and performance of each model in terms of accuracy, sensitivity, specificity, and AUC, and compare our proposed methods with some state-of-the-art MTSC methods. Finally, in Section 8, we discuss the conclusions and future work based on our findings.

## 2. Related Work

Traditional detection methods for TB, including sputum microscopy, culture test, radiology, drug susceptibility testing, whole genome sequencing, and clinical signs/symptoms, have proven effective in acid-fast bacilli detection, point-of-care diagnosis, and cost efficiency. However, these approaches exhibit shortcomings, such as low sensitivity, time consumption, and poor efficacy, which may produce false-negative results, lack of differentiation between various bacterial strains, the inability to detect bacterial viability, and unsuitability for resource-limited settings [12–14]. These limitations may delay TB diagnosis, which may further exacerbate infection severity, raise mortality risk, and enable bacilli transmission in the healthy population. Moreover, erroneous diagnosis can result in imprecise treatment, eventually leading to the development of drug resistance in affected patients [15]. Therefore, the World Health Organization (WHO) has stated that there is an urgent need for a rapid, cost-effective, and sputum-free triage test to detect TB in real-time.

In addition, the importance of developing new diagnostic and detection technologies for the growing number of clinical challenges is rising each year. The analysis of disease-related VOCs represents a new frontier in medical diagnostics due to its noninvasive and inexpensive nature for illness detection. Specific VOC species and their concentration changes for each disease are unique and, thus, make them valuable biomarkers for disease detection [16,17]. Spectrometry and spectroscopy techniques have demonstrated their efficacy in detecting VOCs directly from the headspace of the disease-related cells via urine, blood, skin, or exhaled breath. However, despite their effectiveness, these techniques are often hindered by their high costs, the level of expertise, and the time required to operate the sophisticated equipment necessary for sample analysis [18,19]. To overcome these challenges, some researchers have proposed a novel pathway that enables the use of sensor matrices based on nanomaterials as a clinical and point-of-care diagnostic tool. Nanomaterials have several advantages, including high sensitivity, fast response and recovery time, and synergetic properties when combined. Furthermore, nanomaterial-based sensors can be integrated into portable, low-cost devices through mass manufacturing, enabling noninvasive, easy-to-use, personalized disease diagnosis, and follow-ups. Existing studies [3,16,17] have shown the potential of nanomaterial-based sensors for VOC-based disease diagnosis.

Paper [16] reviewed two complementary approaches to profiling disease-related VOCs by nanomaterial-based sensors: selective and cross-reactive sensing. Our research is based on work [2], where the authors utilized the cross-reactive approach. This method broadly responds to various TB-specific VOCs emitted from the skin's headspace. The VOC selectivity is gained through pattern recognition by obtaining information on the vapor's identity, properties, and concentration exposed to the sensor array.

In [3], the authors reported an artificially intelligent nanoarray for the noninvasive diagnosis and classification of 17 diseases based on exhaled breath VOCs; reference [2] employed a similar type of sensor, consisting of chemiresistive films containing spherical gold nanoparticles (GNPs; core diameter 3–4 nm) capped with different organic ligands, 2D random networks of single-walled carbon nanotubes (RN-SWCNTs) capped with different



organic layers, and polymeric composites. The inorganic nanomaterials within the films are responsible for electric conductivity. In contrast, the organic component provides sites for VOC adsorption. Upon VOC exposure, they are either absorbed onto the sensing surface or diffused into the sensing film, reacting with the organic phase or functional groups that cap the inorganic nanomaterials. This reaction/interaction results in the volume expansion/shrinkage of the nanomaterial film, causing changes in the conductivity between the inorganic nanomaterial blocks. For a sample collection, 40 sensors were employed, each with different functional groups capping the inorganic nanomaterial. This ensured that each sensor yielded a distinct response to individual or patterned VOCs within the sample, generating a full metabolic profile of the tested state, resulting in a pattern of resistance changes detected by the sensor array to a given vapor.

Previous studies primarily concentrated on developing nanomaterial-based sensors for accurately detecting disease-related VOC patterns. However, they did not furnish comprehensive details on the applied classification procedures that procured the results. Furthermore, the dependability and progress of discriminant data classifiers cannot be ensured. Moreover, existing ML approaches are more prevalent in large dataset scenarios. However, approaches such as transfer learning are widely used in small dataset settings, as introduced in Section 1; there is currently a deficiency in similar and extensive datasets that can be utilized as source domains in transfer learning for our task. Therefore, it is critical to develop dependable and suitable ML models pertinent to data-deficient problems that can be combined with other domains and foster their development.

### 3. Background

#### 3.1. Time Series Classification

The rapid expansion of data availability has led to the development of time series classification (TSC) in a wide range of fields, ranging from human recognition [20] and electronic health records [21] to acoustic scene classification [22] and stock market prediction [23]. Thus, TSC has attracted the attention of a large number of researchers. The definition of a TSC task can be categorized into two types:

**Definition 1.** *A univariate time series $X = \{x_1, x_2, \cdots, x_T\}$ is an ordered set of real values with timestamps. The length of X equals the number of real values T.*

**Definition 2.** *A dataset $D = \{(X_1, Y_1), (X_2, Y_2), \cdots, (X_N, Y_N)\}$ consists of a collection of N pairs of $(X_i, Y_i)$, where $(X_i, Y_i)$ is the ith sample and $X_i$ is either a univariate or multivariate time series accompanied by $Y_i$ as its one-hot label vector.*

The TSC task aims to train a classifier over dataset $D$ to map from the time series inputs to a probability distribution over class labels. Our task can be categorized as an instance of the MTSC problem, where each sample comprises a set of MTS inputs (denoted as $X$) and a single corresponding label (represented as $Y$).

#### 3.2. Encoding Time Series as Images by Gramian Angular Field (GAF)

The Gramian angular field (GAF) is one of the most widely used frameworks for encoding univariate time series as 2D images [24]. This approach has recently gained popularity due to its ability to capture cyclical patterns and correlations present in the original time series data, thus enabling researchers to take advantage of the success of deep learning architectures in CV and transfer it into the time series domain. The GAF transformation mainly involves two steps: encoding the univariate time series into polar coordinates and then computing the Gramian matrix of the encoded data.

Before the GAF transformation, the input time series $X = \{x_1, x_2, ..., x_n\}$ first needs to be normalized within the interval $[-1, 1]$ by

$$\tilde{x}_i = \frac{(x_i - max(X) + (x_i - min(X)))}{max(X) - min(X)}.$$ (1)

In the first step, the normalized time series $\tilde{X} = \{\tilde{x}_1, \tilde{x}_2, ..., \tilde{x}_n\}$ of $n$ real-valued time steps can be represented in polar coordinates by encoding the value as the angular cosine and the time stamp as the radius, using

$$\begin{cases} \phi = arccos(\tilde{x}_i), -1 \leq \tilde{x}_i \leq 1, \tilde{x}_i \in \tilde{X}, \\ r = \dfrac{t_i}{N}, t_i \in \mathbb{N}. \end{cases} \tag{2}$$

The equation presented above defines the value of the time stamp $t_i$ and incorporates a constant factor $N$ to regulate the range of the polar coordinate system. In the second step, pairs of angular values from the polar coordinate representation are taken, and their outer products are calculated as follows:

$$G = \begin{pmatrix} \cos(\phi_1 + \phi_1) & \cos(\phi_1 + \phi_2) & \cdots & \cos(\phi_1 + \phi_n) \\ \cos(\phi_2 + \phi_1) & \cos(\phi_2 + \phi_2) & \cdots & \cos(\phi_2 + \phi_n) \\ \vdots & \vdots & \ddots & \vdots \\ \cos(\phi_n + \phi_1) & \cos(\phi_n + \phi_2) & \cdots & \cos(\phi_n + \phi_n) \end{pmatrix} \tag{3}$$

These outer products are then aggregated to form a Gramian matrix, which can be visualized as a 2D image. The resulting GAF image is a compact and information-rich representation of the original time series data. It captures the cyclical patterns and correlations present in the data and allows for the application of a wide range of image-processing techniques for subsequent analysis.

### 3.3. The Long Short-Term Memory (LSTM) Network

A recurrent neural network (RNN) is a neural network that can simulate discrete-time dynamical systems with an input $x_t$, a hidden state $h_t$, and an output $y_t$ [25]. The dynamical systems can be defined by:

$$h_t = f_h(x_t, h_{t-1}) = \tanh(Wh_{t-1} + Ix_t) \tag{4}$$

$$y_t = f_o(h_t) = \text{softmax}(Wh_t) \tag{5}$$

where the subscript t represents the time step. $f_h, f_o$ are the state update function, and output function, respectively. where $f_h$ can use the hyperbolic tangent function $\tanh(\cdot)$, while the output function can usually use the $\text{softmax}(\cdot)$ function that can output a valid probability distribution as the model prediction. $W, I$ represent the recurrent weight matrix and the projection matrix, respectively, which serve as the parameters of the functions.

However, RNNs are often faced with vanishing gradient problems. Long short-term memory is an improved version of RNN [26], incorporating gating functions in the dynamical system to overcome this problem [27]. In a typical LSTM architecture, a memory vector {$\mathbf{m}$, an LSTM hidden state vector $\mathbf{h}$, and the input $\mathbf{x}$ are employed to update the state and generate output at every time step, which can be expressed more precisely according to [28] by

$$\begin{aligned} \mathbf{g}^u &= \sigma(\mathbf{W}^u \mathbf{h}_{t-1} + \mathbf{I}^u \mathbf{x}_t) \\ \mathbf{g}^f &= \sigma(\mathbf{W}^f \mathbf{h}_{t-1} + \mathbf{I}^f \mathbf{x}_t) \\ \mathbf{g}^o &= \sigma(\mathbf{W}^o \mathbf{h}_{t-1} + \mathbf{I}^o \mathbf{x}_t) \\ \mathbf{g}^c &= \tanh(\mathbf{W}^c \mathbf{h}_{t-1} + \mathbf{I}^c \mathbf{x}_t) \\ \mathbf{m}_t &= \mathbf{g}^f \otimes \mathbf{m}_{t-1} + \mathbf{g}^u \otimes \mathbf{g}^c \\ \mathbf{h}_t &= \tanh(\mathbf{g}^o \otimes \mathbf{m}_t) \end{aligned} \tag{6}$$

where $\mathbf{W}, \mathbf{I}, \mathbf{g}$ denote the recurrent weight matrices, projection matrices, and activation vectors, respectively, and the superscripts $\{u, f, o, c\}$ represent the input, forget, output, and the cell state gates. The activation function includes the hyperbolic tangent denoted by tanh

$(\cdot)$, and the logistic sigmoid function $\sigma(\cdot)$. $\otimes$ represents the element-wise multiplication. The computation pipeline for the LSTM network is depicted in Figure 1.

**Figure 1.** The structure of the LSTM network.

### 3.4. The Graph Neural Network Model

Graph neural networks (GNNs) are a general framework for modeling deep neural networks using graph information, such as nodes, edges, and graph structures. The goal with GNNs is to generate representations of nodes based on the graph structure and any feature information of the graph. In recent years, GNNs have become increasingly popular due to their ability to capture complex relationships and dependencies among elements in the graph. It has been successfully applied to various problems, such as node classification, graph classification, etc. In Section 5.4, we first transformed the MTS signal into graph-structured nodes and then employed GNN to classify the node representation.

Spectral Clustering and MinCutPool

Spectral clustering (SC) is a widely-used technique to identify strongly-connected communities within a graph [29]. In the context of GNNs, it can be employed to perform pooling operations that aggregate nodes belonging to the same cluster, which can effectively reduce the dimensionality of the input graph by replacing groups of nodes with a smaller set of nodes, each of which represents a cluster of similar nodes. The new coarsened graph can help to improve computational efficiency while enabling more effective modeling of high-level graph features. However, most SC relies on the eigendecomposition of the graph's Laplacian matrix to project the graph nodes into a lower-dimensional space [30], which can be very expensive, and the result is also graph-specific. Therefore, to overcome the limitations of SC, Bianchi et al. [29] proposed a novel graph clustering method that enables the rapid computation of cluster assignments without the need for spectral decomposition. Then they applied the generated cluster assignment matrix $S$ as an input to the MinCutPool layer, which serves to coarsen the graph by aggregating nodes that belong to the same cluster while preserving the salient features of the original graph.

## 4. Data Preprocessing

### 4.1. Dataset Description

In [2], during the sample collected phase, the study included 928 subjects between the ages of 22 and 60. To establish a robust method for TB detection and eliminate the influence of environmental factors on the samples, the samples and analysis were conducted in three different locations, including New Delhi in India, Cape Town in South Africa, and Riga in Latvia. The study population consisted of 461 healthy controls, including both healthy volunteers and confirmed non-TB samples, denoted by label 1, and 467 newly diagnosed and confirmed pulmonary-active TB patients, each represented by label 0.

As TB involves many bodily systems, it is not easy to be diagnosed with only one unique biomarker when collecting VOCs from the skin's headspace. To overcome the lack of specific biomarkers [31], a combination of 40 non-selective nanomaterial-based sensors is used simultaneously to detect a variety of TB-specific VOCs, providing a comprehensive metabolic assessment of each sample's tested state. Then, each sensor will translate the detected VOCs into resistance signals with a duration of *T* time steps. The raw sensor signal's time length T corresponding to each sample may differ. During the sample measurement, the wearable device is applied directly to the skin on the sample's chest and anterior arm regions (Figure 2). The translated 40 original sensor signals corresponding to one sample are shown in Figure 3, and each raw sensor signal is similar to the one shown in Figure 4. Figure 5 displays the change in the sensor resistivity (i.e., $\Delta R_{end}/R_b$) under different storage conditions, including vacuum, ambient air, and pure nitrogen, respectively, for nine months. As can be seen, the resistivity change in the room air is the largest (35%), followed by the vacuum (19%), and pure nitrogen (17%) [2]. Therefore, it supports the observation of the sensor signal's characteristics in Section 4.2.

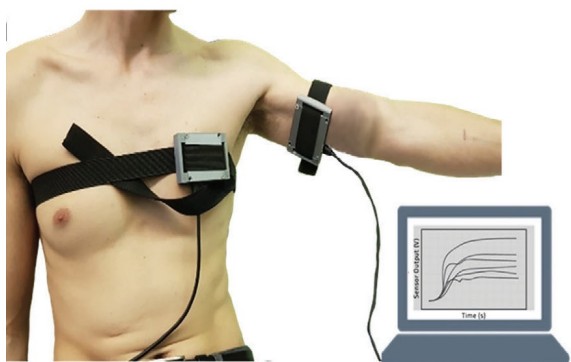

**Figure 2.** Wearable sensor devices on a volunteer's chest and anterior arm.

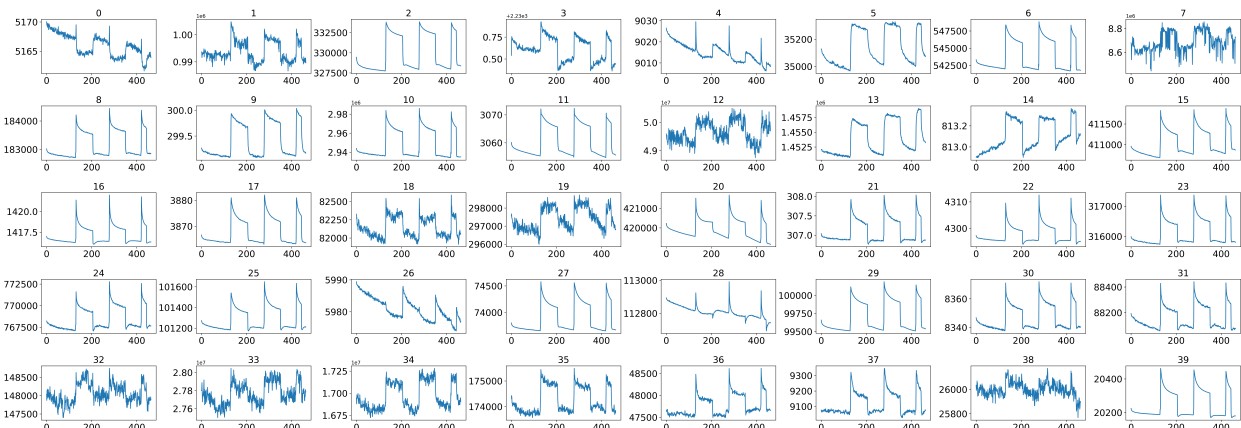

**Figure 3.** The original 40 sensor signals corresponding to one sample before data preprocessing.

The dataset can be represented by $(X, Y)$, where $X \in \mathbb{R}^{N \times n \times T}$ represents the samples, and $Y \in \mathbb{R}^N$ represents the corresponding labels, which are either 0 or 1. Note that $N$ is the number of samples, $n$ denotes the number of sensors (or nodes), and $T$ represents the total time steps of each resistance signal.

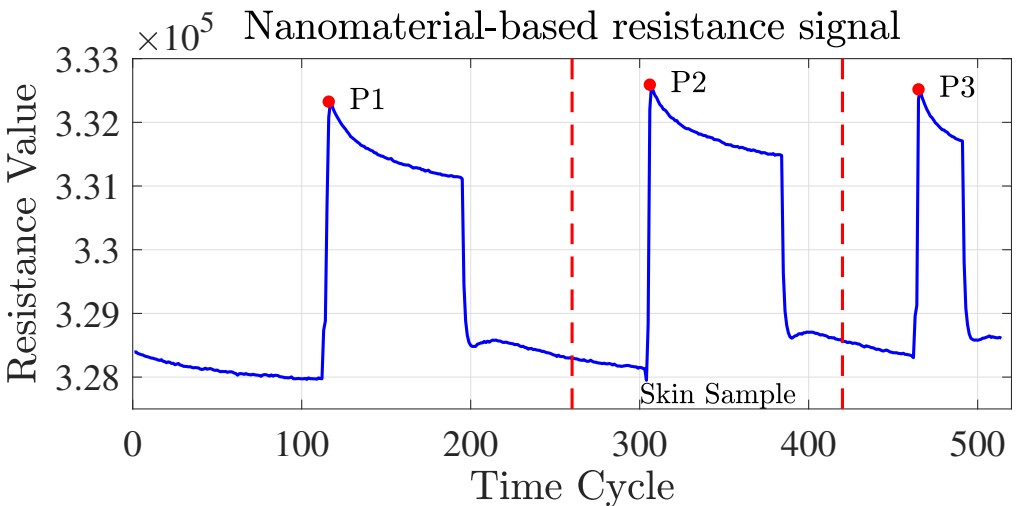

**Figure 4.** Nanomaterial-based sensor resistance signals. Useful signals are obtained under ambient air exposure (within the red dotted line).

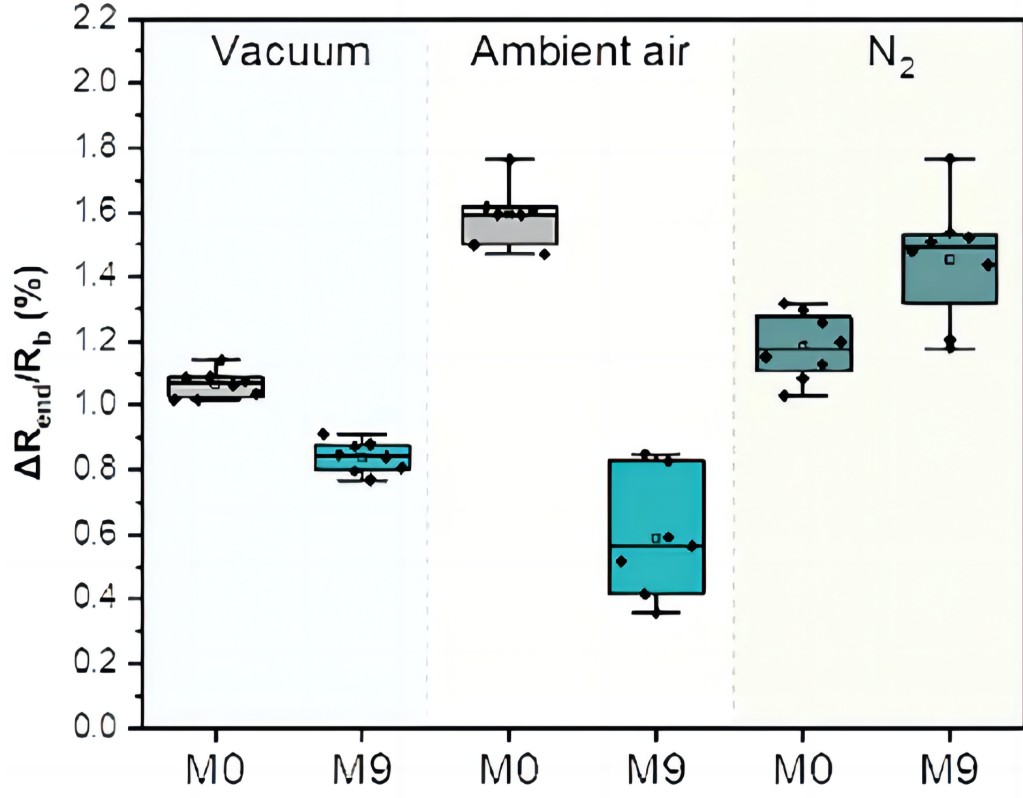

**Figure 5.** The change in the sensor resistivity (i.e., $\Delta R_{end}/R_b$) for different storage conditions at the starting point (M0) and after 9 months (M9).

### 4.2. Middle Part Signal Extraction

The multivariate time series sensor signals were recorded under three different conditions, i.e., vacuum, pure $N_2$, and sample exposure [2] (see Figure 4). The sensor's baseline

responses were recorded for 5 min in a vacuum, 5 min under pure nitrogen (99.999%), 5 min in a vacuum, and 5 min under the sample exposure, followed by a further 3 min under vacuum conditions [2]. Only the resistance signals obtained under the ambient air on skin samples were valuable signals that needed to be extracted from the full signals before applying ML models, which corresponded to the middle peak part of the signal, where the signal curve presented the characteristics of a flat line, a rising peak, and then a return to a flat state.

To achieve this, we first computed the three peak points of the entire signal, denoted by P1, P2, and P3, respectively. Since the length of the middle part of the signal of each sample varied and could fluctuate within a certain range, setting a fixed length in advance and moving P2 or P3 separately to intercept the signal was not feasible. To overcome this problem and obtain the start and end points of the rough middle part signal extracted from each sample, we moved P2 and P3 to the left by 40 and 10 time steps, respectively.

The reason for the different time steps used to move P2 and P3 was to account for the varying lengths observed in the roughly extracted middle part signal across samples. To ensure that the final middle part signal obtained from each sample was of equal length, we first needed to find the minimum length denoted by $l_{min}$ among all extracted signals. We achieved this by shifting P3 to the left by a shorter distance of 10, reserving space for the final signal interception to a consistent length. Accordingly, the final middle part signal of each sample was defined as spanning the starting index and the minimum length by $[\text{start}, \text{start} + l_{min}]$. This was a crucial preprocessing step that helped standardize the signal features and enhance the accuracy of the subsequent analysis.

### 4.3. Data Normalization

Data normalization is one of the essential preprocessing approaches [32]. However, during the signal generation and collection phase, due to different sampling times and the external environments (e.g., pressure, temperature, humidity, etc.) and the characteristics of each sensor itself [2], the resistance values measured by each sensor correspond to each sample having various starting points and large ranges in $O(10^7)$. Therefore, to improve the data quality, we need to normalize the data to the same scale and give each feature a uniform contribution. Inspired by the normalization methods from [32], we use a transformed median normalization method to preserve better relationships in the resistance values; the method is defined as follows:

$$X' = X - \min(X) \tag{7}$$

$$\hat{X} = \frac{X'}{\text{med}(X'_{[0,30]})} \tag{8}$$

where X is the input signal, $\min(\cdot)$ denotes the minimum resistance value from all timestamps, and $\text{med}(\cdot)$ denotes the median value of the input. We first shift each X by its minimum value and obtain $X'$ by Equation (7), and then obtain the normalized signal $\hat{X}$ by dividing $X'$ by the median of its first 30 time steps, as shown in Equation (8). After the data normalization, all signals will have the same minimum point 0, and a smaller scale between $[0, 1]$. This step normalizes each sensor signal into the same range.

### 4.4. Data Calibration

To supervise the sensor's functionality during the experiment and to overcome the possible sensor response drift, making the character of the sample resistance curve align with the original curve of the corresponding sensor, a baseline resistance is measured as a calibration signal before measuring the sample signals. The way we calibrate each sample signal is as follows:

$$\hat{X} = \hat{X} \cdot \frac{\text{Peak}(\hat{C})}{\text{Peak}(\hat{X})} \tag{9}$$

where $\hat{X}$ and $\hat{C}$ are the normalized sample signal and its corresponding normalized calibration signal. We first compute the range ratio between $\hat{X}$ and $\hat{C}$. Since the minimum value for both normalized signals is 0, we only need to consider the peak point of each signal denoted by Peak(·) in this setting. Then each $\hat{X}$ is calibrated by multiplying its corresponding range ratio.

### 4.5. Sensor Selection by Using the Pearson Correlation Coefficient Matrix

Wearable sensors have become more prevalent in settings that require reliability and accuracy, such as in healthcare and clinical diagnosis. Several sensors are usually combined together in order to overcome the relative weaknesses of other sensors, such as sensor uncertainty, limited spatial coverage, imprecision, and malfunctioning [33], to improve the overall accuracy, robustness, and reliability of a decision-making process, and enhance the overall performance of a system [34].

In our case, during the actual sample collection phase, each sample was measured by 40 sensors simultaneously, where some sensors were placed on the subject's anterior arm, while others were placed on the subject's chest, as shown in Figure 2. However, not all 40 sensors have effective and stable signal outputs, as shown in Figure 6. In addition, jagged signal fluctuations can be seen in some sensors that produce noisy and unstable outputs. Therefore, we cannot rely on all of the sensors that were used during measurements, and it is essential to select sensors that can produce stable and clear signal outputs so that we can have a better representation of the entire system, thus improving the model's performance in the subsequent decision-making process.

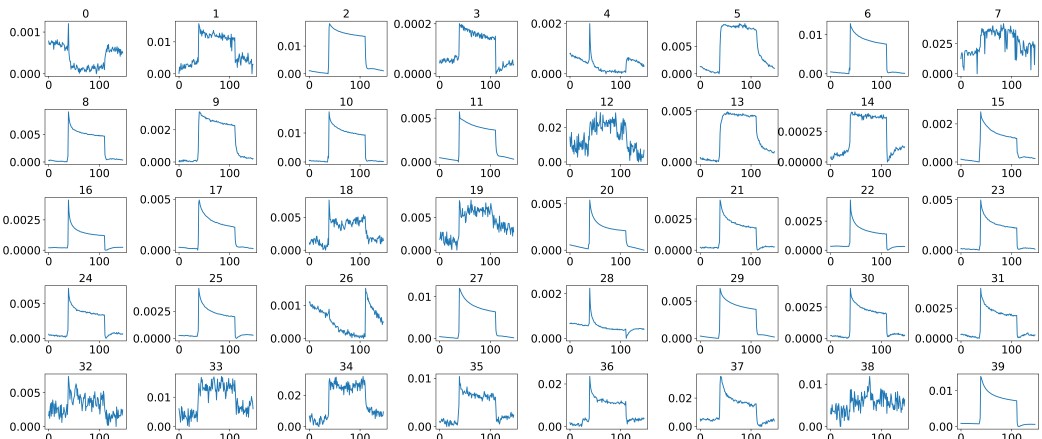

**Figure 6.** Representative 40 sensor signals of a sample after middle-part extraction, where the x-axis of each signal represents the time steps and the y-axis represents the normalized and calibrated resistance values.

An indicator of a bad sensor is that the output signal is volatile, with large fluctuations and sawtooth patterns. Therefore, we use the Pearson correlation coefficient [35] to measure the signal similarities of each sensor in different samples. The similarity coefficient is expected to be small for bad sensors; for good stable sensors, the similarity coefficient will be large. To reduce the possible contingency of the selection and obtain a robust selection result, we aim to avoid excluding sensors that are critical to determining whether a sample is an active TB patient during the decision-making phase.

In the first step, the samples are divided into two groups, one containing active TB patients and the other containing healthy controls. Then ten lists of indices are generated from each group, each with 15 sample indices. Next, we iterate over from the 40 sensors to

compute the Pearson correlation coefficient between different samples randomly selected, corresponding to each sensor, according to:

$$C_{xy} = \frac{\sum_{i=1}^{T}(x_i - \bar{x})(y_i - \bar{y})}{\sqrt{\sum_{i=1}^{T}(x_i - \bar{x})^2 \sum_{i=1}^{T}(y_i - \bar{y})^2}} = \frac{\text{cov}(x, y)}{\sqrt{\text{var}(x) \cdot \text{var}(y)}} \tag{10}$$

where $C_{xy}$ is the Pearson correlation coefficient between signals x and y. Next, the mean coefficient values are calculated from the randomly selected samples to represent the stability of each sensor, where sensors with mean coefficient values lower than 0.65 are classified as non-stable sensors, while the rest are considered good sensors to retain. The Pearson correlation coefficient matrices for both the bad sensors and the good sensors can be seen in Figure 7. In the end, both the active TB and the healthy control groups screened out the same 11 unstable sensors and kept the same 29 good sensors, as shown in Figure 8.

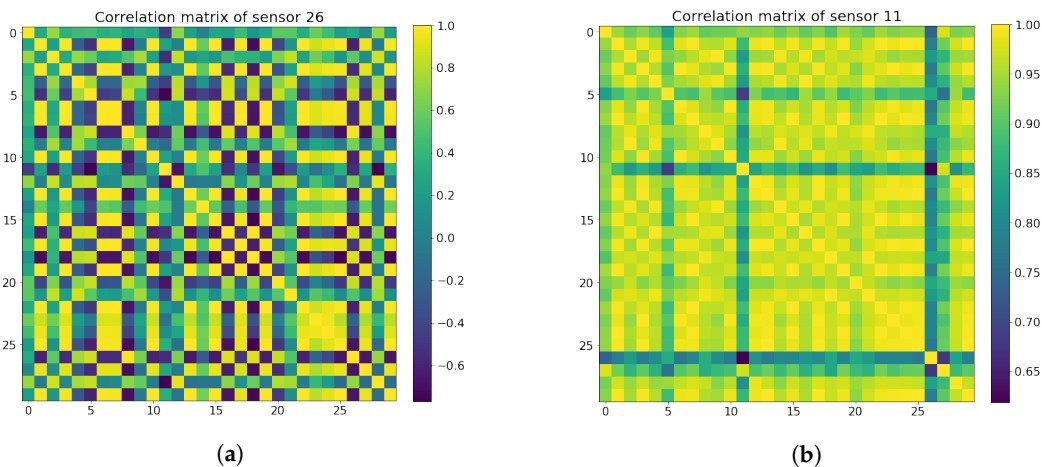

(**a**)                                        (**b**)

**Figure 7.** Pearson correlation coefficient matrix heat maps for both unstable sensors (**a**) and good sensors (**b**). (**a**)Unstable sensor matrix heat map. (**b**) Good sensor matrix heat map.

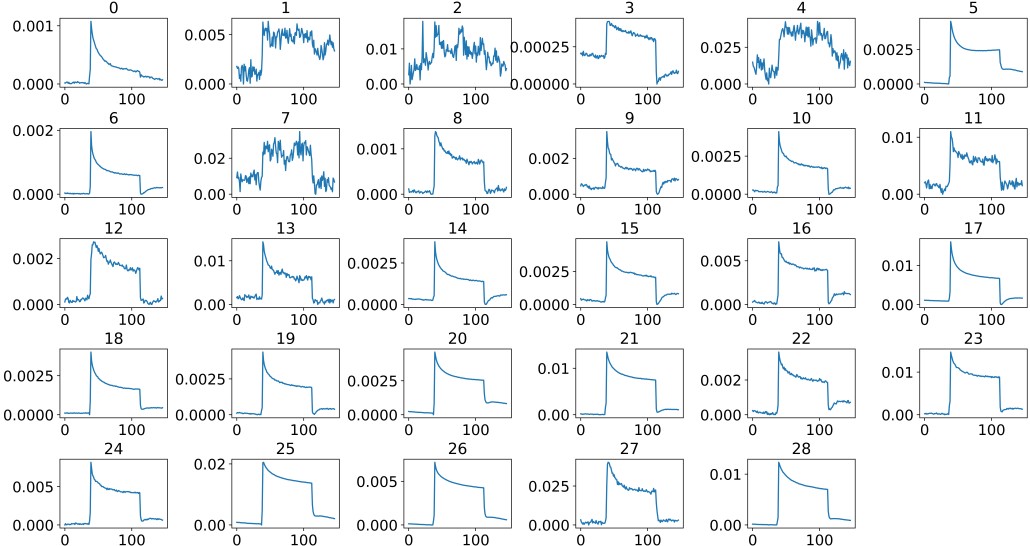

**Figure 8.** Selected 29 sensor signals of a sample after middle-part extraction and stable sensor selection using Pearson's correlation coefficient, where the x-axis of each signal represents the time steps, and the y-axis represents the normalized and calibrated resistance values.

The stability of a signal is mainly attributed to the inherent characteristics of the sensor itself. However, it is essential to note that we cannot rule out the case that when dealing with a different dataset, denoted as $X_b$, a sensor set $S_a$ that is considered stable when

used on a particular dataset $X_a$ may not exhibit the same stability when employed on $X_b$. Therefore, to ensure a stable sensor selection process that generalizes to new data and guarantees accurate subsequent classification, it is advisable to calculate the new sensor stabilization values and choose an alternative set of stable sensors, denoted as $S_b$ when using only $X_b$.

In cases when using both $X_a$ and $X_b$ as input, it is feasible to use an intersection of the stable sensors to achieve reliable classification outcomes. Specifically, $S_c = S_a \cap S_b$ can be used to select the common set of sensors shared between the two datasets.

## 5. Proposed Methods for Multivariate Time Series Classification

### 5.1. Proposed LSTM Network

The TB samples consist of multivariate time series sensor signals, and LSTMs are good at learning temporal dependencies [36]. Thus, we first employ the LSTM network to learn the temporal features from the multivariate time series inputs. The first layer of the LSTM model architecture consists of an LSTM layer that includes 32 LSTM node units. The unfolded internal LSTM network structure is shown in Figure 1. Subsequently, a dropout layer with a dropout rate of 0.2 is applied. Following this, the resulting features are flattened into a vector, which is then forwarded into a dense layer containing 16 units, and activated by the ReLU activation function. The final layer is a single-node output-dense layer, where the predicted labels are generated via the sigmoid activation function. The proposed LSTM model can be seen in Figure 9.

Large complex architectures with large-sized parameters are very prone to having overfitting problems when training with small-sized datasets. Therefore, we only use a small number of units in each layer together with a dropout layer in our model, aiming to avoid or minimize the overfitting problem during the training phase. Finally, a total of 8481 parameters are included in the whole LSTM model. The detailed layer and parameter information is shown in Table 1.

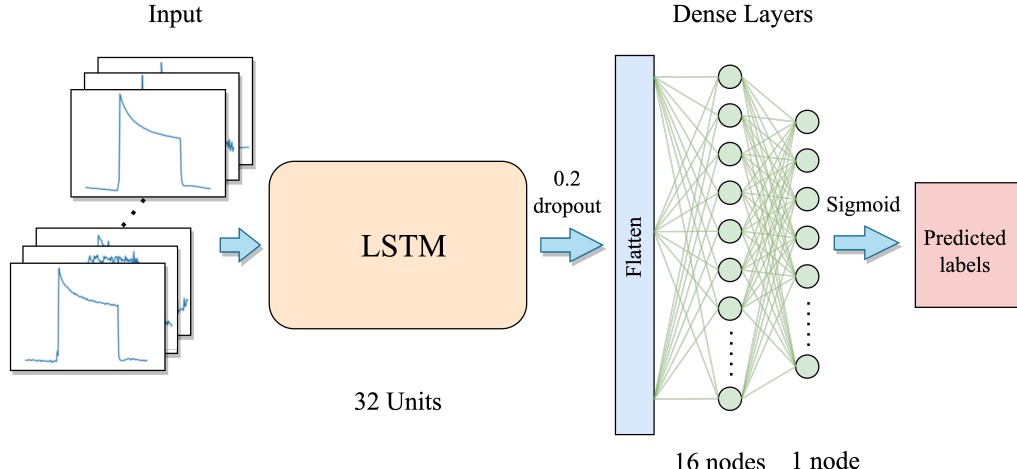

**Figure 9.** The proposed LSTM model architecture.

**Table 1.** LSTM model layers and parameters.

| Layer (Type) | Output Shape | Activation | Parameter Number |
|---|---|---|---|
| lstm_1 (LSTM) | (bs, 32) | – | 7936 |
| dropout_1 (Dropout) | (bs, 32) | – | 0 |
| dense_1 (Dense) | (bs, 16) | ReLU | 528 |
| dense_2 (Dense) | (bs, 1) | Sigmoid | 17 |

During the training phase, we use the Adam optimizer and the binary cross entropy (BCE) function as the loss function, which is a widely used loss function for binary classification problems and can be expressed as:

$$BCE = -\frac{1}{N}\sum_{i=1}^{N}[y_i \cdot \log(p(y_i)) + (1 - y_i) \cdot \log(1 - p(y_i))] \tag{11}$$

where $N$ is the total number of input samples, $y_i$ is the actual label for sample $i$, which is either 0 for active TB patients or 1 for non-TB samples in our case, and $p(y_i)$ is the corresponding predicted probability for the positive class. The BCE loss function provides a useful measure of the discrepancy between the true labels and the predicted probabilities. A lower value of the BCE loss indicates a better fit between the predicted probabilities and the true labels. Conversely, the loss will be large if the predicted probability is far from the true label.

The LSTM model input is $(X, Y)$, where $X \in \mathbb{R}^{N \times n \times T}$ and $Y \in \mathbb{R}^{N}$, the detailed description of the input dataset can refer to Section 4.1, where $T$ equals 147 after the middle part extraction, and $n$ equals 29, which represents the number of stable sensors after the sensor selection.

### 5.2. Convolution Neural Network (CNN)

In recent years, deep learning (DL) has been successfully applied in various domains, including image recognition problems, natural language processing tasks, etc. In light of the tremendous success of DL architectures in these different domains, researchers have begun adopting them for time series analysis as well [37]. Most deep learning-based TSC methods can be divided into two types: generative and discriminative [38]. Generative methods, characterized as model-based methods in the TSC community, are designed to find a suitable time series representation before training a classifier. In contrast, discriminative methods directly learn the mapping between the raw time series and the class probability distributions. The implementation of generative models is more complex than that of discriminative models, while the performance could be better. Thus, the researchers focus primarily on discriminative models, notably on end-to-end approaches for TSC classification tasks [39].

According to the recent comprehensive review of DL-based TSC methods in [37], CNN is the most commonly used structure for TSC tasks due to its robustness and the fact that it requires less training time compared to other DL architectures. Therefore, we propose a CNN architecture to classify the MTS sensor signals. The overall architecture of the proposed CNN is depicted in Figure 10.

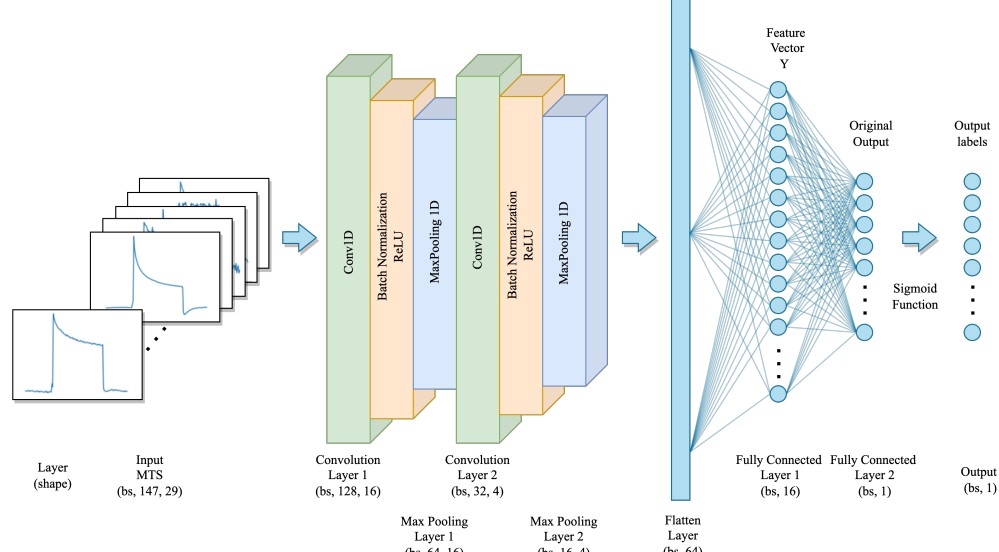

**Figure 10.** The proposed CNN model architecture.

The proposed CNN model takes a set of MTS sensor signals as input, with the shape of $(bs, T, n)$, where $bs$ is the batch size, $T$ equals 147, which represents the time steps of each sensor, and $n$ equals 29, which represents the number of sensors used to correspond to each sample. The proposed model consists of two one-dimensional convolutional layers with different kernel sizes and output channels. Specifically, the first convolutional layer has a kernel size of 20 with 16 output channels, and the second convolutional layer has a kernel size of 2 with 4 output channels. Each CNN layer is followed by batch normalization and a ReLU activation function to model the interactive relationships between multivariate dimensions and the sequential information of the time series. Furthermore, a one-dimensional max-pooling layer with a pooling kernel of size 2 is employed to reduce the spatial dimension of the output from the convolution layer while retaining features with stronger identification. The outputs of the convolutional layers are then flattened into a dense vector and then processed with 2 fully connected layers with 16 units and 1 unit, respectively. Finally, the classification results are obtained by computing the probability of each class by using the sigmoid activation function after the final fully connected layer. The detailed parameter setting of the CNN model is shown in Table 2.

**Table 2.** The parameter settings for the CNN model.

| Layer | Stride | Activation | Kernel Size | Input Shape | Output Shape | Parameter Number |
|---|---|---|---|---|---|---|
| Conv1D_1 | 1 | ReLU | 20 | (bs, 147, 29) | (bs, 128, 16) | 9296 |
| Max Pooling1D_1 | 2 | - | 2 | (bs, 128, 16) | (bs, 64, 16) | 0 |
| Conv1D_2 | 2 | ReLU | 2 | (bs, 64, 16) | (bs, 32, 4) | 2116 |
| Max Pooling1D_2 | 2 | - | 2 | (bs, 32, 4) | (bs, 16, 4) | 0 |
| Flatten | - | - | - | (bs, 16, 4) | (bs, 64) | 0 |
| Dense_1 | - | ReLU | - | (bs, 64) | (bs, 16) | 1040 |
| Dense_2 | - | Sigmoid | - | (bs, 16) | (bs, 1) | 17 |

*5.3. GAF-CNN*

Inspired by [40], we first encoded the univariate time series sensor signals into polar coordinates using the amplitude and phase of the time series. This encoding captures the temporal structure of the time series and allows us to apply the GAF transformation to create the image representation, which was previously introduced in Section 3.2. The transformed images have a fixed shape (batch_size, height, width, channels), where the height and width are equal to the time steps (147), and the input channels are similar to the number of sensors (29 in this case).

The resulting images are then input to the proposed GAF-CNN model which is shown in Figure 11, which consists of two 2D convolutional layers, each followed by a ReLU activation function and a 2D Max Pooling layer. The first and second convolutional layers have 12 and 6 filters, respectively, where the kernel size for both layers is ($5 \times 5$). The 2D max pooling operation uses $2 \times 2$ windows with a stride of 2. After the second max pooling layer, the output is flattened and passed through a fully connected layer with a single neuron and a Sigmoid activation function. The resulting output value is between 0 and 1, representing the probability of the corresponding sample belonging to the positive class. Table 3 shows the model's detailed parameter settings.

**Table 3.** The parameter settings for the GAF-CNN model.

| Layer | Stride | Activation | Kernel Size | Input Shape | Output Shape | Parameter Number |
|---|---|---|---|---|---|---|
| Conv2D_1 | 1 | ReLU | (5, 5) | (bs, 147, 147, 29) | (bs, 143, 143, 12) | 8712 |
| Max Pooling2D_1 | 2 | - | (2, 2) | (bs, 143, 143, 12) | (bs, 71, 71, 12) | 0 |
| Conv2D_2 | 1 | ReLU | (5, 5) | (bs, 71, 71, 12) | (bs, 67, 67,6) | 1806 |
| Max Pooling2D_2 | 2 | - | (2, 2) | (bs, 67, 67, 6) | (bs, 33, 33, 6) | 0 |
| Flatten | - | - | - | (bs, 33, 33, 6) | (bs, 6534) | 0 |
| Dense_1 | - | Sigmoid | - | (bs, 6534) | (bs, 1) | 6535 |

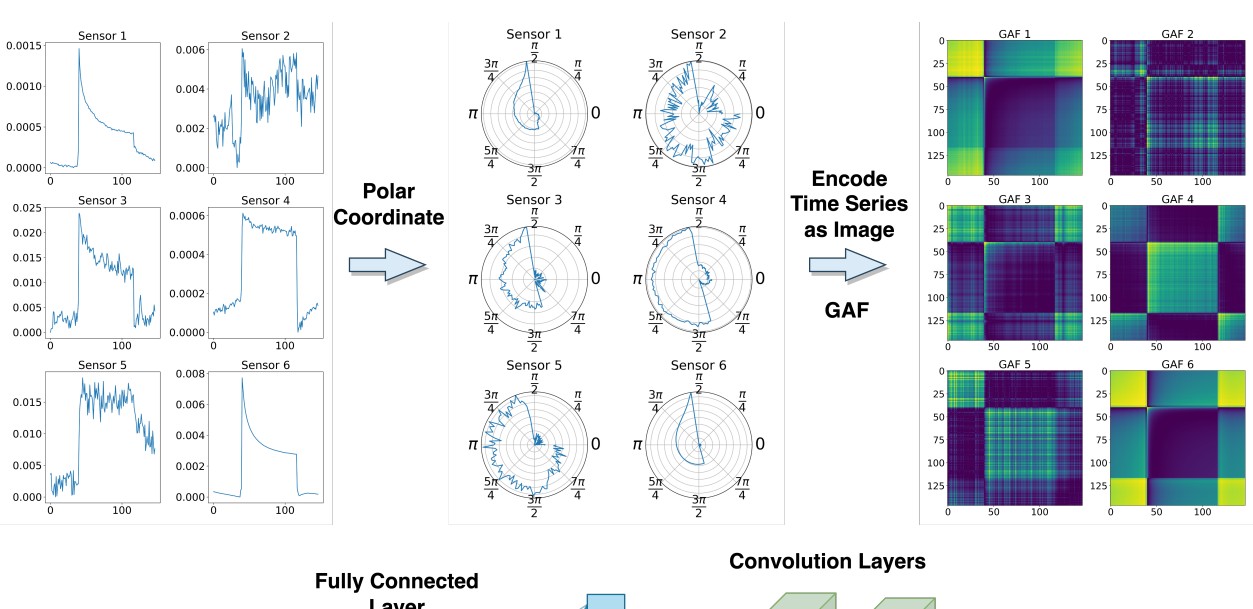

**Figure 11.** GAF-CNN model architecture.

*5.4. MTSC with Graph Laplacian and MinCutPool*

In [41], a novel graph pooling-based framework multivariate time series classification with variational graph pooling (MTPool) is proposed to obtain an expressive global representation of MTS. This study is one of the few approaches that use GNN to solve the MTSC problem from a graph perspective. In contrast, others solve MTSC problems using transformer-based frameworks (e.g., [42]), ensemble methods that ensemble over several univariate classifiers independently and then aggregate the predictions from each classifier to generate a single probability distribution for each TSC task (e.g., [43]), and variants of recurrent neural networks (e.g., multivariate LSTM fully convolutional network (MLSTM-FCN) [44]).

This section proposes the MT-MinCutPool framework (Figure 12), a modified MTPool to solve the MTSC task. The main differences are that we employ the graph's Laplacian matrix to construct the graph and use the MinCutPool to cluster similar nodes within a graph into one cluster to coarsen the graph. This section mainly contains five parts: graph structure learning using the Laplacian matrix, temporal feature extraction, spatial–temporal modeling, graph coarsening by MinCutPool, and graph-level embedding classification.

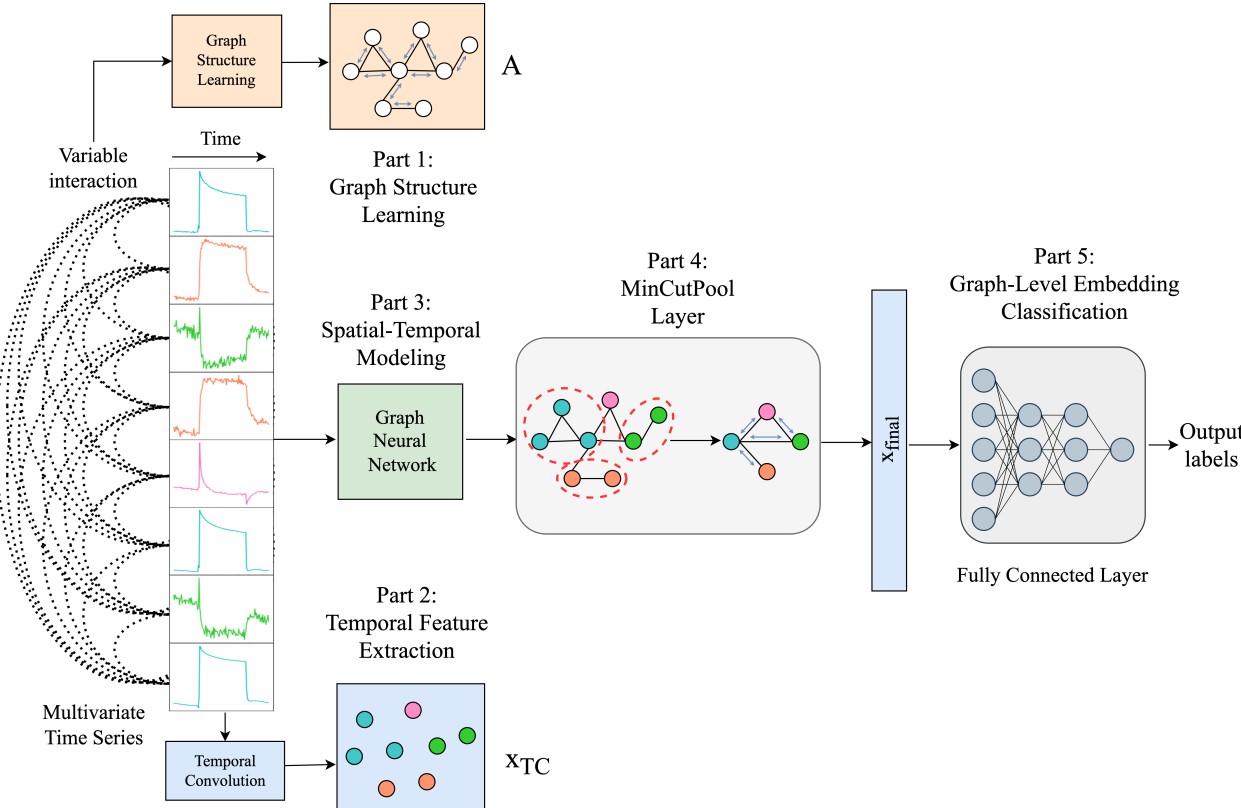

**Figure 12.** MT-MinCutPool model architecture.

Graph Structure Learning Using Laplacian Matrix

Graph-based representations and algorithms for handling structured data rely heavily on constructing meaningful graphs. Dong et al. [45] addressed the problem of learning the graph Laplacian, which is equivalent to learning graph topologies, such that the input data will be transformed into graph signals with smooth variations in the resulting topology. Therefore, we also use the graph Laplacian matrix to represent the graph structure in the first part of the framework.

The dynamic time-warping method (DTW) will likely produce a more reliable similarity assessment between two time series than other distance measurement methods, such as Euclidean distance, which matches timestamps regardless of feature values. For input samples $X = \{x_1, x_2, \cdots, x_N\} \in \mathbb{R}^{n \times T}$, $N, n, T$ represent the number of samples, sensors, and total timestamps, respectively. We first use the DTW method to calculate the distance matrix $DTW \in \mathbb{R}^{N \times n \times n}$ between each sensor within MTS samples. Then we will construct an adjacency matrix $A$ that can be used to calculate the following degree matrix $D$ and the Laplacian matrix $L$. In the meantime, to improve the model's overall training efficiency, enhance the model's robustness, and reduce the impact of the noise introduced by the sensors, a threshold $\theta$ is set to make the distance matrix $DTW$ sparse:

$$A_{ij} = \begin{cases} 1, & \text{if DTW[i, j]} < \theta, \\ 0, & \text{if DTW[i, j]} > \theta. \end{cases} \tag{12}$$

After having the adjacency matrix $A \in \mathbb{R}^{N \times n \times n}$, the elements of each row in matrix $A$ are first added up to obtain vector $a \in \mathbb{R}^{N \times n}$, which is then diagonalized to generate degree matrix $D$, and finally, after, having the adjacency matrix $A$ and the degree matrix $D$. By definition, the graph Laplacian matrix $L$ can be obtained through $D - A$. The detailed process of constructing the graph Laplacian matrix is shown in Algorithm 1.

---

**Algorithm 1:** Build the Laplacian adjacency matrix.

```
// N: # samples, n:  # sensors, T: # timestamps
```
**Input:** $X \in \mathbb{R}^{N \times n \times T}$, $\theta$
**Output:** $L \in \mathbb{R}^{N \times n \times n}$
```
// (1) Build Distance Matrix DTW
```
DTW $\leftarrow$ empty matrix with shape $\mathbb{R}^{N \times n \times T}$
**for** $i = 1$ **to** $N$ **do**
    x $\leftarrow$ X $[i]$
    distance $\leftarrow$ empty matrix with shape $\mathbb{R}^{n \times n}$
    **for** $j = 1$ **to** $n$ **do**
        **for** $k = 1$ **to** $n$ **do**
```
                // dtw (·) is the dynamic time warping distance function
```
            distance $[j, k] \leftarrow$ dtw (x $[j]$, x $[k]$)
        **end**
    **end**
    DTW $[i] \leftarrow$ distance
**end**
```
// (2) Build Degree Matrix D
```
D $\leftarrow$ empty array with shape $\mathbb{R}^{N \times n \times T}$
A $\leftarrow$ int (DTW $< \theta$)
**for** $i = 1$ **to** $N$ **do**
```
    // diagonal (·) is the diagonalized function
```
    adj $\leftarrow$ A $[i]$
    D $[i] \leftarrow$ diagonal $(\sum_{k=1}^{n}$ adj $[k, :])$
**end**
```
// (3) Build Laplacian Matrix L
```
L $= D - A$
**return** L

---

### 5.4.1. Temporal Feature Extraction

The purpose of temporal convolution is to extract features along the time axis, as well as to design a temporal feature matrix $X_{TC} \in \mathbb{R}^{N \times n \times d}$, where $d$ is the dimension of the new extracted features, which can serve as a strong reference for the subsequent classification step. When analyzing time series data, it is essential to consider both numerical values and long-term patterns. Therefore, to extract features from the time dimension and keep as much of the origin pattern as possible, we adopt the method employed in the prior work as presented in [41], which employs multiple convolutional neural network channels with varying kernel sizes:

$$X_{TC} = ||_{i=1}^{m} f_i = ||_{i=1}^{m} \sigma(W_i * X + b) \tag{13}$$

where $||_{i=1}^{m}$ denotes the concatenation operation that merges the feature maps generated by the first to the $m$-th CNN filters, where subscript $i = \{1, 2, \cdots, m\}$ represents the specific CNN filter number. Each $f_i \in \mathbb{R}^{N \times n \times d_i}$ is the output tensor of each convolution layer containing the extracted temporal feature. Moreover, it is given by the convolution of the input tensor $X$ with the learnable filter $W_i \in \mathbb{R}^{out \times in \times ks}$, where *out* is the number of output channels, *in* is the number of input channels, *ks* is the kernel size of the filter, followed by an element-wise bias addition $b \in \mathbb{R}^{out}$, and an activation function $\sigma(\cdot)$, such as ReLU$(\cdot)$, to introduce non-linearity into the model. The operator $*$ denotes the convolution operation.

After applying the convolutional layer with a given kernel size $ks$, the new sequence length can be computed according to $d_i = (T - ks)/s + 1$, where $ks$ and $s$ are the kernel size and stride step from the learnable filter $W$, respectively. Finally, we obtain the temporal feature matrix $X_{TC} \in \mathbb{R}^{N \times n \times d_{TC}}$ by concatenating each convolved tensor $f_i$, where $d_{TC} = \sum_{i=1}^{m} d_i$ is the new sequence dimension. The concatenation of the extracted temporal features from various time scales provides a reliable reference for the subsequent classification task.

### 5.4.2. Spatial–Temporal Modeling

Spatial–temporal modeling is an essential task in many applications, such as skeleton-based action recognition [46], traffic forecasting [47], etc. Graph neural networks (GNNs) have demonstrated promising results on spatial–temporal modeling tasks. Their ability to directly apply filters on the graph nodes and their neighbors enables the model to learn representations that capture both the spatial dependencies and the temporal patterns of the data.

Graph convolution networks (GCNs) are specific types of GNNs designed to deal with graph-structured data. Therefore, in this part, we adopt GCN to operate on the input graph-structured data, typically represented as an adjacency matrix, $A$, which indicates whether an edge connects two nodes, and a feature matrix, $X$, which contains the features of each node in the graph. The graph convolution operation can be defined as:

$$\tilde{X} = G(A, X_{TC}, W, b) = \sigma(A * X_{TC} * W + b) \tag{14}$$

where $\tilde{X}$ is the output feature matrix. $G(\cdot)$ is the graph convolution function, which takes the adjacency matrix $A$, the input feature matrix $X_{TC}$, a learnable weight matrix $W$ of the convolutional filter, and a bias term $b$ as input.

In the graph convolutional layer, the feature matrix $X_{TC} \in \mathbb{R}^{N \times n \times d_{TC}}$ is first multiplied by the adjacency matrix $A \in \mathbb{R}^{N \times n \times n}$, which in our case is the graph Laplacian matrix, it allows the information to be propagated from every single node to its neighbors in the graph. Then the output is linearly transformed by a weight matrix $W \in \mathbb{R}^{N \times n \times d_{out}}$ through another multiplication. After adding the bias term $b$, the new feature matrix is transformed using a nonlinear activation function, such as ReLU or tanh, where $\sigma(\cdot)$ is the activation function. Moreover, we finally obtain the resulting output $\tilde{X} \in \mathbb{R}^{N \times n \times d_{out}}$. The graph convolution process updates the graph's node representation and the output node features $\tilde{X}$ can be used for the subsequent graph classification task.

### 5.4.3. Graph Coarsening by MinCutPool

To reduce the computational complexity of the GNN while preserving its expressive power, we combine the clustering method with the MinCutPool pooling method proposed in [29] in this part of our framework to coarsen the graph.

Let $\tilde{A} = D^{-\frac{1}{2}} A D^{-\frac{1}{2}} \in \mathbb{R}^{N \times n \times n}$ be the symmetrically normalized adjacency matrix, where $D$ is the degree matrix. The cluster assignment matrix $S$ is computed using a multilayer perceptron (MLP) activated by a softmax function to map each node feature $x_i$ of the $i$-th row of matrix $S$. Specifically, it can be expressed as

$$\bar{X} = GNN(\tilde{X}, \tilde{A}, W_{GNN}) \tag{15}$$

$$S = MLP(\bar{X}, W_{MLP}) \tag{16}$$

where $\tilde{X}$ is the matrix of the node representation generated from the previous graph convolution layer, and $\bar{X}$ is the new feature matrix yielded by one or more subsequent graph convolution layers. $W_{GNN}$ and $W_{MLP}$ are the trainable parameters. Then the cluster assignment matrix $S$ and the normalized adjacency matrix $\tilde{A}$ are fed into the MinCutPool

layer to obtain the pooling node representations of the coarsened graph. The pooling process is computed as follows:

$$A_{pool} = S^T \tilde{A} S; \quad X_{pool} = S^T X \qquad (17)$$

where the entry $x_{j,k}^{pool}$ from $X_i^{pool} \in \mathbb{R}^{C \times d}$ is the sum of feature $k$ among the items in cluster $j$ from sample $i$, which is weighted by the cluster assignment scores from $S$, where $i = \{1, \cdots, N\}$ is the sample index, $C$ is the number of clusters, and $d$ is the new feature-length sequence after the graph convolution. $A_i^{pool} \in \mathbb{R}^{C \times C}$ is the coarsened symmetrical adjacency matrix, where the matrix element $a_{j,k}$ is the weighted sum of the edges between cluster $j$ and $k$. The entries $a_{j,j}$ are the weighted sums of the node edges within a cluster. Each MinCutPool layer will generate a cut loss term $\mathcal{L}_u$ [29], and the weight parameters $W_{GNN}$ and $W_{MLP}$ are optimized by minimizing $\mathcal{L}_u$ during training, thereby increasing the likelihood of clustering similar nodes together.

### 5.4.4. Graph-Level Embedding Classification

After obtaining the graph-level embedding from the MinCutPool layer, the resulting output graph representation $X_{pool} \in \mathbb{R}^{N \times C \times d}$ is flattened into a vector form, referred to as $X_{dense}^{N \times (C*d)}$. Subsequently, $X_{dense}$ is forwarded to the succeeding fully connected layers for the final classification.

## 6. Experiments
### 6.1. Evaluation Metrics

To evaluate the performance of our proposed models with some state-of-the-art methods for the MTSC problem, we utilize several widely used binary classification metrics, including:

- Accuracy: It measures the proportion of the correct predictions among all of the predictions made by the models.
- Sensitivity (true positive rate, TPR): It measures the proportion of true positive models made by the model out of all actual positive samples. It indicates the ability of the model to correctly identify the positive samples, which is very important for clinical settings. It indicates the ability of a model to identify negative samples correctly.
- Specificity (true negative rate, TNR): It measures the proportion of true negative predictions made by the model out of all actual negative samples.
- AUC: (Area under the ROC curve): It considers the performance of a classifier over all possible threshold values, taking into account both the true positive rate (sensitivity) and the false positive rate (1—specificity).

The following formulas can express the above metrics:

$$\text{accuracy} = \frac{TP + TN}{TP + TN + FP + FN} \qquad (18)$$

$$\text{sensitivity} = \frac{TP}{TP + FN} \qquad (19)$$

$$\text{specificity} = \frac{TN}{TN + FP} \qquad (20)$$

where TP, TN, FP, and FN represent true positive, true negative, false positive, and false negative, respectively. A higher value indicates better performance for all of the evaluation metrics, while a low value indicates poor performance.

### 6.2. Experimental Setup

During the experiment phase, the dataset $(X, Y)$ was partitioned into training, validation, and test sets to ensure a reliable evaluation of the model's performance. Specifically, 80% of the data was used for training and validation, while the remaining 20% was reserved

for testing. This enabled the model to be trained and tuned on a subset of the data, with the held-out test set serving as an unbiased measure of its performance on unseen data. We only had a total of 928 samples, which was a relatively small dataset for ML classification tasks. To efficiently use the existing data to train the model and better represent each model's performance, we adopted the widely used technique of stratified cross-validation for model training and evaluation. Cross-validation presents numerous advantages over conventional methods, including mitigating overfitting issues, particularly prevalent in scenarios involving small datasets, improved data utilization, a robust performance evaluation, and the capability for hyperparameter tuning. In our experiment, we employed a rigorous evaluation process to ensure the reliability of our results. Firstly, we set the number of folds to 5. Then during each training iteration, the model was trained on four folds, i.e., 80% of the train–validation dataset. At the same time, the remaining held-out fold, i.e., 20% of the train–validation dataset, was reserved for validation. This process was repeated for each fold, with a different fold held out as the validation set each time. After training each fold with a single model, we saved and loaded the models and performed five-fold cross-validation on the training and validation sets again. Each fold was evaluated using its corresponding trained model. We then selected the best-performing model based on its accuracy on the validation dataset and assessed it on the unseen test dataset using various metrics, including accuracy, sensitivity, specificity, and the AUC score. To further increase the robustness of our evaluation, we repeated this process ten times, each using a unique random seed to split the training and validation dataset. Finally, we computed the mean test accuracies, sensitivities, specificities, and AUCs across the ten rounds to obtain the final performance of each model. This thorough evaluation process allowed us to ensure the reliability of our results and provide a more comprehensive understanding of the performance of each model. The proposed LSTM, CNN, MT-MinCutPool, and GAF-CNN were trained with 2000, 1000, 500, and 500 epochs, respectively, with a learning rate of $1 \times 10^{-5}$, $5 \times 10^{-5}$, $1 \times 10^{-6}$, and $1 \times 10^{-6}$. In all cases, the early stopping patience was set to 50, and a batch size of 32 was used. Each model's parameters were updated throughout the training process using the BCE loss and the stochastic gradient descent. In the case of the MT-MinCutPool model, in addition to the BCE loss, the min-cut loss, which was generated from the graph coarsen process in Section 5.4.3, was incorporated as an additional component of the overall loss function.

## 7. Results and Discussion

### 7.1. Main Results

The evaluation results of the proposed models and the baseline models are presented in Tables 4 and 5, respectively. The mean performance of each evaluation metric for both the proposed and the baseline models are calculated and displayed in the last column of each table. We highlighted the highest test scores for each metric in bold for both the baseline models and the proposed models.

**Table 4.** Train and validation accuracy, sensitivity, specificity, and AUC values for the baseline models.

|  |  | LSTM | CNN | GAF-CNN | MT-MinCutPool | Model Mean |
|---|---|---|---|---|---|---|
| **Accuracy** | train | 0.646 | 0.916 | 0.659 | 0.688 | – |
|  | valid | 0.69 | 0.687 | 0.692 | 0.69 | – |
|  | test | 0.611 | 0.606 | **0.639** | 0.604 | 0.615 |
| **Sensitivity** | train | 0.618 | 0.931 | 0.766 | 0.76 | – |
|  | valid | 0.675 | 0.715 | 0.78 | 0.756 | – |
|  | test | 0.631 | 0.694 | **0.777** | 0.728 | 0.71 |
| **Specificity** | train | 0.676 | 0.901 | 0.544 | 0.612 | – |
|  | valid | 0.704 | 0.658 | 0.597 | 0.619 | – |
|  | test | **0.594** | 0.533 | 0.532 | 0.5 | 0.54 |
| **AUC** | test | 0.634 | 0.657 | **0.692** | 0.661 | 0.661 |

Our experimental findings reveal that the mean performances of our proposed models outperform those of the state-of-the-art baseline approaches for all evaluation metrics. Specifically, the average accuracy, sensitivity, specificity, and AUC score of the proposed models are all higher than those of the baseline models. GAF-CNN demonstrates the best overall performance among all proposed models, achieving the highest test accuracy, sensitivity, and AUC score. The LSTM model exhibits the highest specificity. The GAF-attention model performs the best among all baseline models, with the highest accuracy, specificity, and AUC score. Additionally, the MLSTM-FCN model attains the highest sensitivity score. GAF-CNN has the highest accuracy of 0.639 and the highest sensitivity of 0.777 among all models. In contrast, the GAF-attention model has the highest specificity of 0.637 and the highest AUC score of 0.695.

**Table 5.** Train and validation accuracy, sensitivity, specificity, and AUC values for the baseline models.

|  |  | **MTPool [41]** | **GAF-Attention [40]** | **MLSTM-FCN [44]** | **Model Mean** |
|---|---|---|---|---|---|
| **Accuracy** | train | 0.563 | 0.755 | 0.733 | − |
|  | valid | 0.577 | 0.663 | 0.693 | − |
|  | test | 0.517 | **0.631** | 0.586 | 0.578 |
| **Sensitivity** | train | 0.682 | 0.78 | 0.757 | − |
|  | valid | 0.683 | 0.688 | 0.733 | − |
|  | test | 0.655 | 0.622 | **0.664** | 0.647 |
| **Specificity** | train | 0.436 | 0.728 | 0.709 | − |
|  | valid | 0.464 | 0.636 | 0.651 | − |
|  | test | 0.4 | **0.637** | 0.521 | 0.519 |
| **AUC** | test | 0.538 | **0.695** | 0.648 | 0.627 |

We employed the AUC-ROC curve to visualize each model's performance better. The ROC is a probability curve that plots the true positive rate (TPR) against the false positive rate (FPR), and AUC represents the degree of each model's separability. The higher the AUC, the better the model distinguishes between positive TB patients and healthy controls. We can see from Figure 13 that the order of the model performance from high to low (in terms of AUC-ROC) is as follows: GAF-Attention > GAF-CNN > MT-MinCutPool > CNN > MLSTM-FCN > LSTM > MTPool.

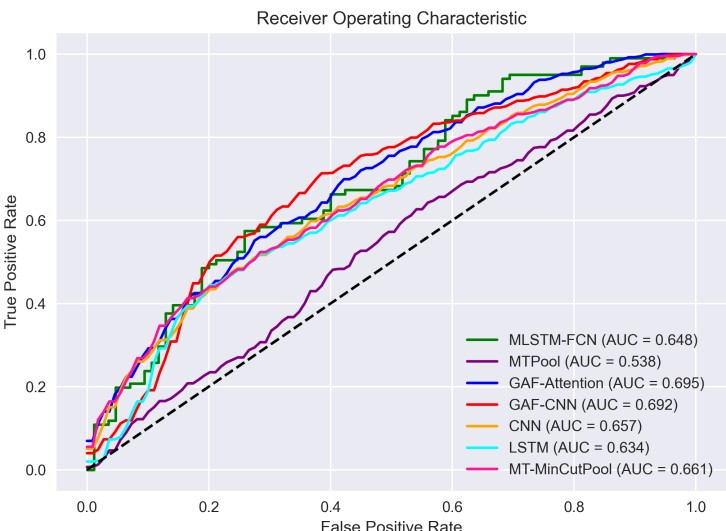

**Figure 13.** The receiver operating characteristic (ROC) curve of each model.

The training and validation accuracy and loss curves for each selected model with the highest accuracy score on the validation dataset are shown in Figures 14–17.

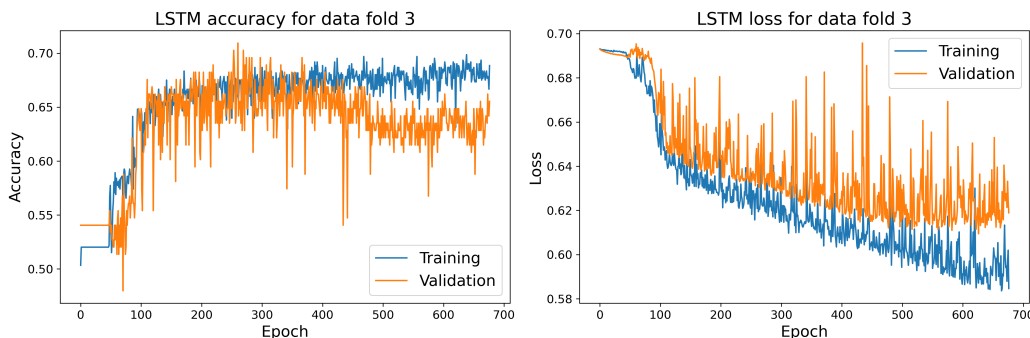

**Figure 14.** The training and validation accuracy and loss curve for the best LSTM model.

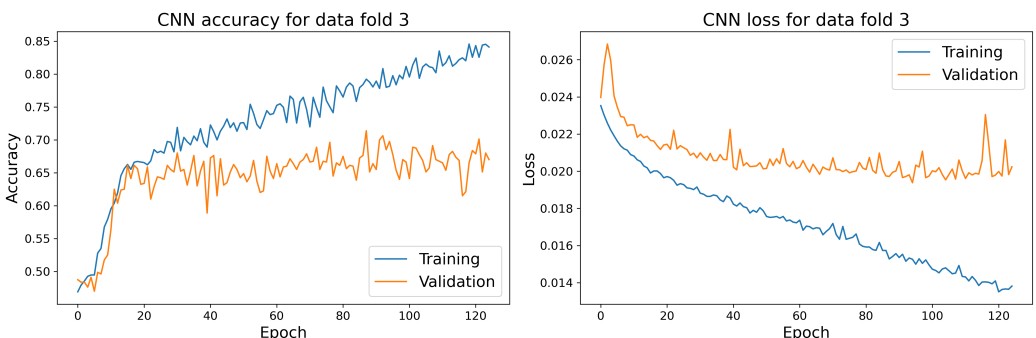

**Figure 15.** The training and validation accuracy and loss curve for the best CNN model.

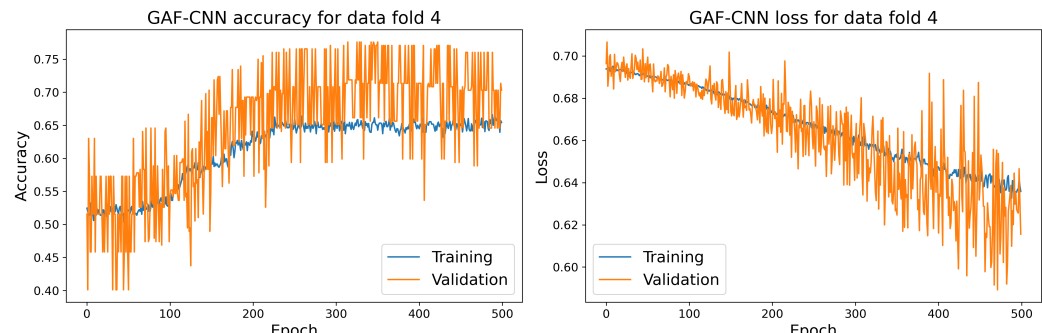

**Figure 16.** The training and validation accuracy and loss curve for the best GAF-CNN model.

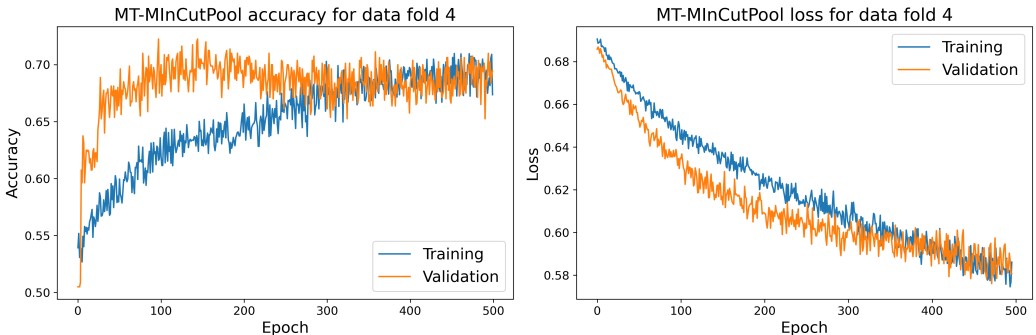

**Figure 17.** The training and validation accuracy and loss curve for the best MT-MinCutPool model.

### 7.2. Discussion

The MT-MinCutPool model is modified according to the MTPool model. It performs relatively well compared to the proposed LSTM and CNN models since it has higher sensitivity and AUC scores. Furthermore, its accuracy, sensitivity, specificity, and AUC

scores are all higher than those obtained using the MTPool model. This might be due to the following two reasons. Firstly, in MT-MinCutPool, we use the Laplacian matrix to represent the graph topology, which contains more graph information than the correlation coefficient matrix employed in MTPool. Secondly, MinCutPool is designed to preserve important information by retaining highly connected nodes in the graph. Unlike other widely-used graph pooling approaches, such as GraphSAGE [48], which learns to aggregate feature information from each node's nearby neighbors, MinCutPool directly identifies similar nodes that are strongly connected and aggregates them into one cluster. This approach ensures that important information is preserved during the pooling process. Thus, by employing the graph Laplacian to build the graph structure and using MinCutPool to replace the adaptive pooling layer in MTPool, we obtain a better performance than the MTPool model.

To better adapt to small datasets in our experiment, the proposed CNN and LSTM are designed to be lightweight models, with fewer layers, in order to extract adequate input features. As a result, both models showed an overall superior performance in comparison to the average performance of the baseline models.

Furthermore, the results of our proposed GAF-CNN model demonstrate superior performance compared to the baseline GAF-Attention model in terms of accuracy and sensitivity. Specifically, GAF-CNN achieved an accuracy of 0.639 and the highest sensitivity of 0.777 among all models, while the GAF-Attention model achieved an accuracy of 0.631 and a sensitivity of 0.622. However, the GAF-attention model had the highest specificity of 0.637 among all models, and its AUC, i.e., 0.695, was slightly higher than that of GAF-CNN, i.e., 0.692. These results suggest that both models have their unique strengths and limitations and may be more suitable for different applications depending on the specific requirements of the task.

The effectiveness of using GAF image conversion and the GAF-CNN makes us believe that these methods can be extended to a wider range of contexts and similar applications beyond our immediate tuberculosis MTSC task. Specifically, we suggest a further investigation into the following potential areas:

1.  Medical diagnosis: GAF image conversion could further assist in identifying a range of medical conditions, from electrocardiogram (ECG) rhythms to Alzheimer's signals. GAF image conversion in these fields may allow for better clinical decision-making and enhance the accuracy of machine learning models.
2.  Financial time-series analysis: GAF image conversion could be leveraged to predict stock prices and fluctuations in the currency market and further enhance the effectiveness of ML algorithms in predicting trends and changes.
3.  Speech recognition: GAF-based image classification could enable more accurate identification of speech patterns, phonemes, and speech denoising. The use of images could be particularly effective in noisy environments where traditional audio inputs may be challenged.

In conclusion, converting time series into images has excellent potential; further experimentation and research in these suggested areas are recommended to explore the full potential of this method.

## 8. Conclusions

In this study, we propose several ML-based approaches, including LSTM, CNN, GAF-CNN, and MT-MinCutPool, to tackle the problem of TB classification. The TB dataset comprises multivariate time series sensor signals and a small dataset with 928 samples. We first demonstrated a standard pipeline of some data preprocessing approaches. Subsequently, we evaluated the proposed models on the TB dataset and compared the results with several state-of-the-art baseline methods. Our proposed methods exhibit better overall performance compared to the baseline models. Our observations show that basic and lightweight models perform better than complex models, which are more appropriate for large dataset scenarios. The proposed MT-MinCutPool model outperformed MTPool in all

aspects. Thus, it is a viable and effective model for multivariate time series signals, even for larger datasets. Our work is focused on the TB dataset classification. Therefore, one of the limitations of our work is that the proposed and baseline models were solely evaluated on this particular TB dataset. Future work may involve extending our proposed models to other similar tasks in the field of multivariate time series classification.

**Author Contributions:** Conceptualization, I.C. and C.L.; methodology, I.C. and C.L.; software, C.L.; validation, C.L.; formal analysis, C.L.; investigation, I.C. and C.L.; resources, R.V. and H.H.; data curation, R.V. and C.L.; writing—original draft preparation, C.L.; writing—review and editing, I.C., C.L., H.H. and R.V.; visualization, C.L.; supervision, I.C.; project administration, I.C. and H.H.; funding acquisition, H.H. All authors have read and agreed to the published version of the manuscript.

**Funding:** This work was supported by the Phase-II Grand Challenges Explorations award of the Bill and Melinda Gates Foundation (grant ID: OPP1109493) and Horizon 2020 ICT grant under the A-Patch project (grant ID: 824270).

**Institutional Review Board Statement:** The clinical trials received ethical approvals from the Ethical Committees of the respective hospitals: AIIMS, New Delhi: IEC/NP-103/13.03.2015, RP-39/2015, and the University of Cape Town: 307/2014.

**Informed Consent Statement:** Informed consent was obtained from all subjects involved in the study.

**Data Availability Statement:** The data that support the findings of this study are available from the corresponding author upon reasonable request. An open-source version of our work is accessible on 5 May 2023 at: https://github.com/ChenxiLiu6/TB-Classification.git.

**Conflicts of Interest:** The authors declare no conflict of interest. The funders had no role in the design of the study; in the collection, analyses, or interpretation of data; in the writing of the manuscript; or in the decision to publish the results.

## Abbreviations

The following abbreviations are used in this manuscript:

| | |
|---|---|
| BCE | binary cross entropy |
| CNN | convolution neural network |
| CV | computer vision |
| DL | deep learning |
| DTW | dynamic time warping |
| GAF | Gramian angular field |
| GCN | graph convolution network |
| GNN | graph neural network |
| LSTM | long short-term memory |
| MTF | Markov transition field |
| ML | Machine Learning |
| MLSTM-FCN | multivariate LSTM fully convolutional network |
| MT-MinCutPool | multivariate time series with MinCutPool |
| MTPool | multivariate time series classification with variational graph pooling |
| MTS | multivariate time series |
| MTSC | multivariate time series classification |
| NLP | natural language processing |
| NNs | neural networks |
| SC | spectral clustering |
| TB | tuberculosis |
| TSC | time series classification |
| VOC | volatile organic compound |
| WHO | World Health Organization |

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
