# Peer review of "Nanomaterial-Based Sensor Array Signal Processing and Tuberculosis Classification Using Machine Learning"

_jlpea, doi:10.3390/jlpea13020039_

Round 1
Reviewer 1 Report
Given the type of scientific journal to which this paper has been submitted, it would be worth considering whether the description of the sensor used and the electronics associated with it should have been described more fully. Although another publication that contains a description of the sensor and that is written by one of the authors (Rotem Vishinkin) has been referenced in the paper under review, not even in this referenced work does the sensor or its electronics be sufficiently described.
-Subsections 3.1 and 3.2 need to be explained and justified more clearly. Much essential information is missing to correctly define the protocol followed to extract the signals from the patients. It is also important to describe in greater depth and clarity the characteristics of the electrical signals recorded by the sensors.
-Lines 198 and 199 of the text are confusing and do not justify the need for the shift operation.
-References [6] and [23], which describe the previous work with the sensor used in this work, are the same reference.
Reviewer 2 Report
This article proposes to address the problem of insufficient dataset when developing classification models for tuberculosis(TB), using nanomaterial-based sensor array signal processing and machine learning. In their proposed solution, they have used LSTM, CNN, Gramian Angular Field-CNN, and Multivariate Time Series with MinCutPool to classify a small TB dataset of multivariate time series sensor signals. They have evaluated and validated the proposed method by comparing their obtained results with those of the commonly used state-of-the-art models. Their reported results show that lightweight models are more appropriate for small dataset problems such as TB classification problems. The results also show that the Multivariate Time Series with MinCutPool outperforms the baseline MTPool.
This is a well-written and well-presented article; the problem has been clearly defined. The proposed methods well explained, the validation and results well-presented and interpreted.
Major concerns
· The major weakness is that they ought to have had a related work section, although some related works have been mentioned in the introduction section, however, the authors have spent a great deal of efforts presenting background concepts of machine learning algorithms.
Minor concerns
· Figure 15 needs to be closer to where it has been referenced within the text, i.e., before the discussion section on page 18.
· In section 3.2, why shifting indices p2 and p3 to the left by 40 and 10 respectively?
Figure 4 should be referenced before figure 5. Otherwise, you may have to change the figure number.
Reviewer 3 Report
The authors present a comparative analysis of supervised learning approaches to identify tuberculosis from nanomaterial-based sensor data. The tested algorithms are presented with a high degree of explanatory clarity and the information reported allows for the reproducibility of the approaches.
I propose a number of small comments further to improve the presentation of an already high-quality work:
- I suggest the authors reorganize the content in the introduction. I would start by introducing the context, i.e. the diagnosis of tuberculosis, and then talk about the potential of Machine Learning.
- Can the use of time-series conversion into images be used in other application contexts? The authors can suggest in which other contexts the proposed classification approaches can be used.
Reviewer 4 Report
Review: Nanomaterial-Based Sensor Array Signal Processing and Tuberculosis Classification Using Machine Learning
In this manuscript, Liu and coauthors apply several data preprocessing methods and deep learning architectures to build multivariate time series classification models for detecting tuberculosis with sensor signals of VOC from wearable devices. These proposed models show great accuracy and specificity improvement over baseline models and demonstrate their capability for MTSC tasks.
The topic in this manuscript of using various machine learning models for disease diagnosis is of great interest to researchers in machine learning. It clearly explains method structures, and the training and evaluation experiments, but some important explanations and justifications are missing. And as the dataset used in this work is too small, which makes the evaluation of models questionable. Overall, it deserves publication in JLPEA after the authors are given a chance to consider the following comments:
1. In Page 2, 4th paragraph, can the authors give more explanation of volatile organic compounds VOC signal data? What is detected with these sensors? What is the difference in data from these 40 sensors? How do these sensors translate VOCs to resistance signals? There is only VOC from healthy people and ones with tuberculosis. Is it possible that people with other diseases show similar VOC?
2. It would be better if the authors could provide more justification for using the models CNN and LSTM. CNN is normally used for data with spatial structure, however, in line 317 to line 320, the input has dimensions of time steps and sensor channels, as such, using proximity based on the sensor channel orders doesn’t make sense. For LSTM, it is unclear the temporal dependence of these VOC signals, can the authors add more explanation?
3. After data preprocessing, data from 29 sensors out of 40 are used, can a different dataset give different stable sensors? Can the authors comment on how to make it generalize to new data, like what if a used sensor here is unstable in another dataset?
4. In section 5.2 (line 497), for the data partition, can the author consider leaving out data of patients and healthy people from certain regions in the test set, would it help to improve the generalization of the models?
5. Would the authors consider adding a penalty on the model parameters to the loss function to avoid overfitting?
6. Can the authors add comments on how to improve the model interpretability and make the diagnosis prediction results more reliable and truthful?
7. In Figure 4, can the authors add more details to the caption, what is the x and y axis here?
The font size is too small for Figures 5, 6, 9, and 12. In Lines 218 and 219, there are typos in the equation labels. Can the authors also add the GitHub repo link to the data availability section?
